# Investigation of the Joining Technology of FRP/AZ31B Magnesium Alloy by Welding and Riveting Hybrid Bonding Method

**DOI:** 10.3390/ma12132167

**Published:** 2019-07-05

**Authors:** Hongyang Wang, Nan Li, Liming Liu

**Affiliations:** School of material science and engineering, Dalian University of Technology, No.2 Linggong Road, Ganjingzi District, Dalian 116024, China

**Keywords:** AZ31B Mg alloy, fiber reinforced polymer, dissimilar materials, welding and riveting hybrid joining technology, laser-arc hybrid welding source

## Abstract

A novel joining technology was applied to join Fiber Reinforced Polymer (FRP) and AZ31B Mg alloy, which combined the laser-arc welding source and riveting joining methods. The design idea of the stepped rivet was proposed. The weld morphology, mechanical properties, microstructures of welds under two different rivet structures were investigated. FRP and AZ31B Mg could be joined successfully by the new hybrid joining method when it used two different structural rivets. The maximum tensile shear load of the joint under stepped rivet of small size was only 800 N, while that of the joint under stepped rivet of the larger size could reach 1419 N, nearly 90% of that of FRP. There was no reaction between the FRP plate and AZ31B rivet. While the magnesium elements and aluminum elements diffused and reacted with other elements in the FRP plate/AZ31B plate interface.

## 1. Introduction

Due to environmental pollution and resources shortage, reducing energy consumption and carbon emissions has become the future development goals of manufacturing industries. To achieve energy conservation and emission reduction, vehicle manufacturers are urged to satisfy the requirements of current economic and environmental policies [1].

Having a lightweight automobile is one of the most effective means of reducing carbon dioxide emissions and fuel consumption, and in automotive lightweight technology, the use of lightweight materials is the focus of research [2,3]. Light alloys, such as magnesium, aluminum, titanium, are the most frequently used in automotive lightweighting. Among them, magnesium alloy is the lightest metal material in industrial applications and has excellent mechanical properties, high specific strength and specific stiffness, which is an ideal lightweight material for the new energy vehicles [4].

Research on the dissimilar materials welding technology of magnesium alloys and aluminum alloys, magnesium alloys and steel, magnesium alloys, and titanium alloys has received extensive attention [5,6,7,8,9,10,11]. The research on Fiber Reinforced Polymer (FRP) composites has become a popular topic for lightweight materials due to their advantages of wear resistance, corrosion resistance, high strength, low density, and so on [12,13]. With the in-depth application of composite materials in the automotive and aerospace fields, the joining technology of magnesium alloys and resin-based composite materials will gradually attract more scholars’ attention. FRPs are inevitably joined to lightweight alloys because of the extensive use of a variety of lightweight materials, but the physical and chemical properties of them are obviously different, and the traditional melted welding methods are not fit for the joining of FRP composite and the light alloys.

The common methods of joining FRP to metals are adhesive bonding and mechanical fastening [14,15,16]. However, due to the stress concentration of mechanical fastening and the difficulty in predicting the failure time of adhesive bonding, the application of adhesive melting and riveting quickly reach their technical limit in the Mg and polymer joining process [17]. M. Wahba et al. [18] explored the feasibility of laser welding of polyethylene terephthalate resin (PET) and AZ91D magnesium alloy. They proved that the strength of the metal-side laser-irradiation joints was higher, exceeding 2600 N. K.W. Jung et al. [19] studied the laser direct joining of Carbon Fiber Reinforced Polymer(CFRP) CFRP to zinc-coated steel and investigated the feasibilities, characteristics, and mechanisms of dissimilar joints between CFRP and zinc-coated steel. However, the properties of the welding joints still needed to be improved. Therefore, scholars have introduced a series of hybrid joining technologies in combination with riveting, bonding, and welding techniques [20,21,22,23].

Goushegir et al. [24] used friction spot welding technology to join AA2024 with CFRP and investigated the parameters, which influenced the mechanical properties of the joint. G. Meschut [25] proposed the resistance element welding-hybrid joining method, which combines welding with adhesive bonding to achieve a favorable joining between steel 22MnB5 and CFRP, but the heat transfer leads to failure of the adhesive in a certain radius around the joint. L. Blaga et al. [26] proved and investigated the feasibility of FricRiveting on PEI-GF/Titanium/Aluminum connections, and lap shear strengths of the welds were up to about 200 MPa. And we used the welding and riveting hybrid bonding method to join the Al alloy and polyether-ether-ketone (PEEK) PEEK composites successfully, and the interface structure between the Al alloy and the composite is the key factor for the property [27]. The hybrid bonding technology has a significant technical advantage in the joining of different materials [23] and polymer.

The study of the joining method between Mg alloy and FRP is an important part for the lightweighting manufacture, and the welding and riveting hybrid joining method is a new hybrid joining technology, which is suitable for joining lightweight materials and FRP. In this paper, FRP and AZ31B magnesium alloy are taken as the research objects. The influences of two different rivet structures on the weld morphology, mechanical properties, and microstructure are discussed separately. The elements distribution and reaction phase on the interface are analyzed by EPMA and XPS, to understand the joining mechanism between the FRP and Mg alloy.

## 2. Materials and Method

FRP (100 mm × 30 mm × 3 mm) and AZ31B magnesium alloy (100 mm × 30 mm × 1.5 mm) were selected as base materials in this paper. The chemical composition of the AZ31B Mg used is listed in Table 1. The base material of the FRP is the polyether-ether-ketone thermoplastic.

To reduce the thermal effect on the composite resin matrix during the welding process, the rivet with a stepped structure was used in this experiment. The structure of magnesium step rivet is shown in Figure 1. To compare the influence of the rivet structures on the welding process, two structures rivets were used in the experiment, such as R1 = 2 mm, R2 = 4 mm, R3 = 6 mm (rivet I) and R1 = 3 mm, R2 = 4 mm, R3 = 6 mm (rivet II). Before welding, the surface oil on the AZ31B magnesium alloy was removed by sandpaper and steel brush, then the metal surface was washed away with acetone.

The schematic diagram of the hybrid welding process is shown in Figure 2. The specimens of the two processed materials were overlapped, and the center of the two holes coincided. The magnesium alloy plate was on the top of the FRP plate, and the lap area was 30 mm × 30 mm. Then, the magnesium rivet passed through the two through-holes from bottom to top, securing both plates. The sample was placed on a continuously variable welding positioner platform with the center of the rivet coinciding with the center of the platform. The specimens were clamped by a fixture to keep it level. The welding platform made a uniform circular motion, and the sample rotated with the platform. The sample was kept at a uniform circular motion, and the laser-arc hybrid welding source above the sample remained stationary during the welding process. First, the welding source was aligned at the edge of the rivet on the surface of the sample. Then, the sample was welded until a complete circular weld was done. The hybrid welding parameters are shown in Table 2. The parameters of the pulse laser were as follows: 1.064 μm wavelength, 120 mm core diameter, 0 mm defocusing distance. During the welding experiment, 99.99% pure argon was used as a shielding gas from the TIG torch. In the paper, a pulsed Nd:YAG laser with a maximum average power of 1000 W and a 70 A-tungsten inert gas (TIG) electrical arc were used as hybrid heat sources. A laser-TIG welding source can control the shape of the molten pool well, which can reduce the pool width while ensuring the penetration.

The AZ31B magnesium alloy base metal is based on a α-Mg solid solution, and the black β-Mg_17_Al_12_ precipitate phase is dispersed at the grain boundary. Due to the dynamic recrystallization after rolling, the microstructure is equiaxed with uneven size. The microstructures of AZ31B magnesium alloy and FRP are shown in Figure 3. The appearance and cross sections of the joints with two different rivets were observed by an optical microscope (OM, MEF4, Leica, Germany). The size of specimens for tensile-shear testing was 170 mm × 30 mm. The microstructures were analyzed by a scanning electron microscope (SEM, SUPRA55, Zeiss, Germany). The element distributions in the Mg plate/FRP interface and the Mg rivet/FRP interface were detected by the electron probe micro-analyzer (EPMA, EPMA-1600, Shimadzu, Japan), and compounds identification was investigated by X-ray photoelectron spectroscopy (XPS, ESCALAB™ 250Xi, ThermoFisher, Waltham, MA, USA).

The schematic diagram of specimens for tensile-shear testing is shown in Figure 4. The lap shear tests are conducted by using a universal tensile testing machine with a fixed strain rate of 1.0 mm/s. The ASTM D5868-01 sub-standard sized are considered for preparing tensile specimens.

## 3. Results and Discussion

### 3.1. Weld Seam and Property

The morphologies and cross sections of the joints with two different rivets are shown in Figure 5a–d. Figure 5a shows the morphology of the weld using rivet I. A smooth appearance of the weld was obtained. The upper surface of the rivet I was melted, and there were no defects, such as excessive penetration and porosity. Figure 5b shows the morphology of the weld using rivet II. Only the surface of the rivet was partially melted, which was near the magnesium plate. The welding appearance of the first joint (rivet I) is wider than the second one (rivet II).

Figure 5c,d show the cross-sections of the welded joint using rivet I and rivet II, respectively. Both weld joints were typical laser-arc hybrid welded joints. Laser-arc hybrid welding process has deeper penetration and higher speed, which can precisely control the welding energy. Therefore, the welds formed are deep-melt welds. By contrast, both had the characteristics of wide and narrow weld seams.

The tensile curve of the FRP plate with a hole is shown in Figure 6a, and the maximum peak load is 1500 N. The tensile curves of the rivet I joint and the rivet II joint are shown in Figure 6b,c, respectively. The maximum tensile shear load of the rivet I joint was 800 N, which was only about 53% of the GFRP base material, and the maximum load of the rivet II joint was 1419 N, which was 90% of the GFRP. It can be inferred that the force form of rivet I was similar with the rivet II joint during the stretching process The forms of the force of the pullout fracture mode and the brittle fracture mode of the joints are pretty similar.

The rivet I joint was fractured at the weld area between the rivet and the plate, while the rivet II joint was fractured on the FRP. It is mainly related to the heat distribution during the welding process. The rivet structure changes the heat transfer in the welding process. The diameter of the first step plane of the rivet I was small, and the rotating welding platform led to a prominent overlap in the heat source area. Although the penetration can be effectively increased by the laser beam, the increase of the laser energy directly caused thermal damage to the FRP, due to the overlap of the heat source action regions. The first step plane diameter of the rivet II was slightly larger than that of the rivet I, and there was no heat source overlap phenomenon. Thus the heat can be transferred to the depth direction. A favorable butt joint was formed between the first step of the rivet II and the plate, and a better lap joint was formed between the second step plane and the plate. The composite welded structure of butt and overlap in the rivet II joint effectively improved the overall performance of the welded joint. The diagram of the bonding structure of the joint is shown in Figure 7.

### 3.2. Microstructure of the Hybrid Bonding Joint 

Table 3 shows the microstructure of the rivet I joint and rivet II joint. Table 3 presents the microstructure of the welding joint, the heat affected zone on the Mg plate, and the heat affected zone on the rivet, respectively. It was observed that the grains in the heat-affected zones were slightly larger than that of the base metal. Since laser-arc hybrid welding speed is fast, and the heat input is small, so the heat-affected zone is narrow. It was found that the hardness of the heat affected zone was lower than that of the base metal.

The size in the fusion zone was relatively smaller compared with the base metal. The molten pool cooled and solidified rapidly, and the crystal grains could not grow up.

Region A of Figure 5c is an important area for the bonding joint. SEM observation of region A in Figure 5c is shown in Figure 8a. As shown in Figure 8a, it can be observed that there is a small crack, and some FRP material can be found in the gap between the second step of the rivet and Mg plate. It is formed by melting and gasification of the FRP plate approaching the welding source. Figure 8b is the SEM magnification image of the region D in Figure 8a, where three materials are bordered. Figure 8b shows that the molten composite enters the gap between the Mg rivet and the Mg plate. It can be found that the CFRP was melted during the welding process, the melted CFRP flew into the gap between Mg plate and Mg rivet.

### 3.3. The Bonding Mechanism of Interface between the Mg and FRP

The bonding mechanism of the interface between the Mg alloy and the FRP is very important for the property of the joint. Thus the bonding mechanism between the element Mg, C, and O was discussed individually. As the bonding structure of rivet I and rivet II joint is nearly the same, the EPMA and XPS analysis of the interface was only studied in the rivet II joint. Figure 9 and Figure 10 are electron probing analysis diagrams (EPMA) of the region B and region C, respectively, in Figure 5b. Region B is the interface between the Mg plate and the FRP plate (interface I) near the heat source, and the C region is the interface between the Mg rivet and the FRP plate (interface II). As shown in Figure 9a,b and Figure 10a,b, the distribution of Si elements were also large where the Al elements were concentrated in the FRP plate. 

The Mg plate/FRP plate interface was tightly combined, where there were no obvious cracks. Some Al elements and Mg elements were diffused from the Mg plate to the FRP plate. In the Mg rivet/FRP plate interface, some fine cracks were found and the two were not combined, and the reinforcing fibers and the resin matrix did not change significantly. It meant that the resin was in the liquid state in the welding process, and the Mg alloy was in the solid state.

The results of interface I by X-ray photoelectron spectroscopy analysis (XPS) are shown in Figure 11. Figure 11a is the analysis result of Mg elements. The presence of Mg oxides and MgCO_3_ indicated that Mg elements might be oxidized during the welding process. Figure 11b,c show the results of the analysis of O elements and C elements, respectively. The O elements existed in the form of metal carbonates and organic C–O, respectively, and the C elements existed in the form of C–C, C–O and C=O in the interface I, which demonstrated the possibility of the presence of O–C=O. The analysis results of Al elements and Si elements are shown in Figure 11d,e. They were both shown in the form of aluminosilicate and proved the existence of aluminosilicate. And Al elements also existed as alumina, showing that Al elements might be oxidized. The diagram of the interfaces is shown in Figure 12.

## 4. Conclusions

In this paper, FRP and AZ31B Mg alloy were well joined by the novel welding and riveting hybrid joining method. The conclusions of this work are as follows:

The appearances of welds obtained by using two different rivet structures were perfect. The load tensile test results of the two joints showed that the rivet I joint represented the pullout fracture mode, and the rivet II joint was the brittle fracture mode. The maximum tensile shear load of the rivet I was 800 N, while that of the rivet II was 1419 N, which was 90% of the FRP. These results showed that different rivet structures affected the overall performance of the welded joint.

The microstructure observation showed that the grain size of the heat affected zone of the joint was larger than that of the base metal, and the internal grains of the weld were equiaxed. The results of SEM showed that when welding with rivet I, the FRP portion near the heat source melted and gasified, which caused cracks in the weld, explaining why the tensile strength of the welded joint was too low. The results of EPMA and XPS showed that there might be Mg oxide, Al oxide, and aluminosilicate in the Mg plate/FRP plate interface, which was helpful for the improvement of the dissimilar materials joint.

## Figures and Tables

**Figure 1 materials-12-02167-f001:**
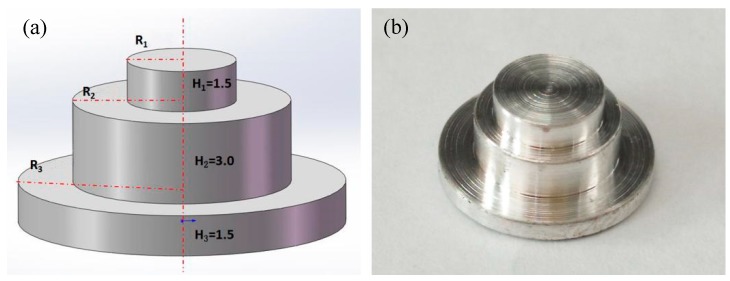
The structure of magnesium step rivet. (**a**) structure of the rivet; (**b**) Mg rivet.

**Figure 2 materials-12-02167-f002:**
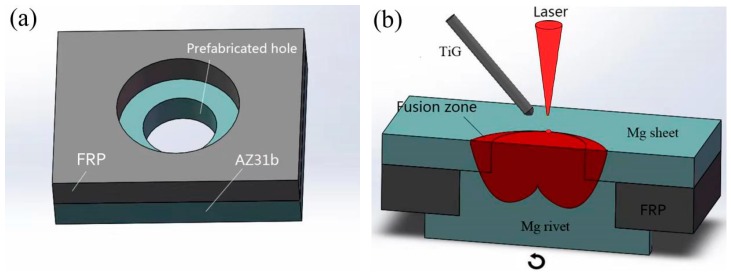
The schematic diagram of the hybrid welding process. (**a**) prefabricated hole in Mg and Fiber Reinforced Polymer (FRP) and (**b**) laser-TIG hybrid welding process.

**Figure 3 materials-12-02167-f003:**
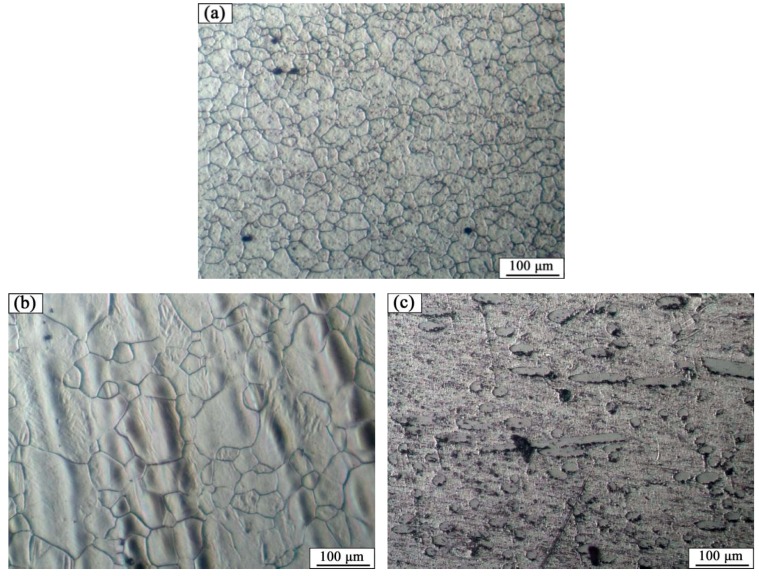
The microstructures of base materials (**a**) AZ31B magnesium alloy plate, (**b**) AZ31B magnesium alloy rivet, and (**c**) FRP plate.

**Figure 4 materials-12-02167-f004:**
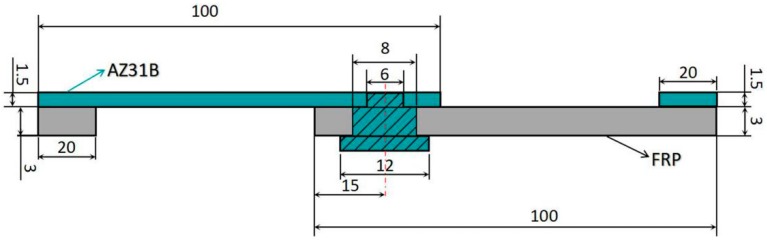
Schematic diagram of the specimen for tensile-shear testing (mm).

**Figure 5 materials-12-02167-f005:**
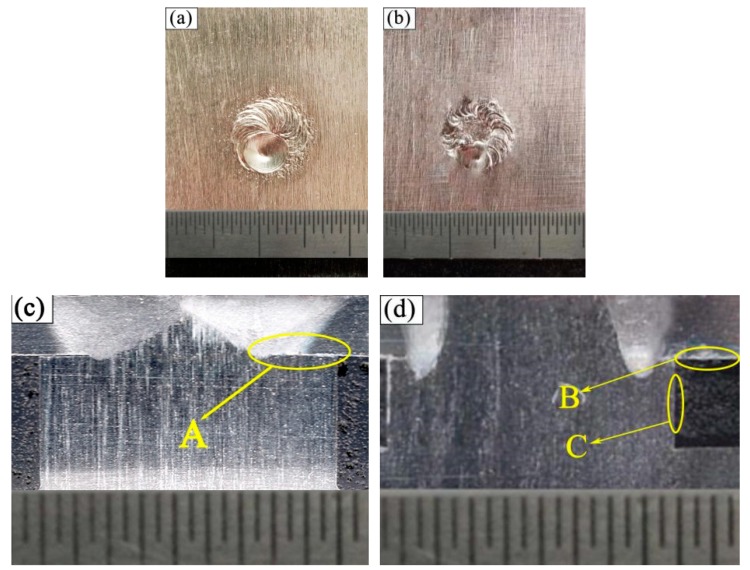
The morphologies and cross sections of the joints using different rivet. (**a**) the morphology of the weld using rivet I, (**b**) the morphology of the weld using rivet II, (**c**) the cross-section of the welded joint using rivet I, and (**d**) the cross-section of the welded joint using rivet II (mm).

**Figure 6 materials-12-02167-f006:**
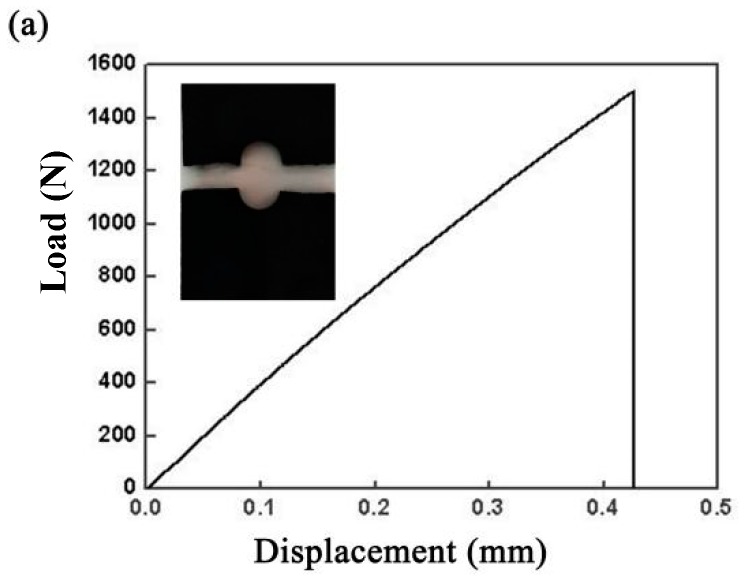
Tensile curves of (**a**) the FRP plate with a hole, (**b**) the rivet I joint, and (**c**) the rivet II joint.

**Figure 7 materials-12-02167-f007:**
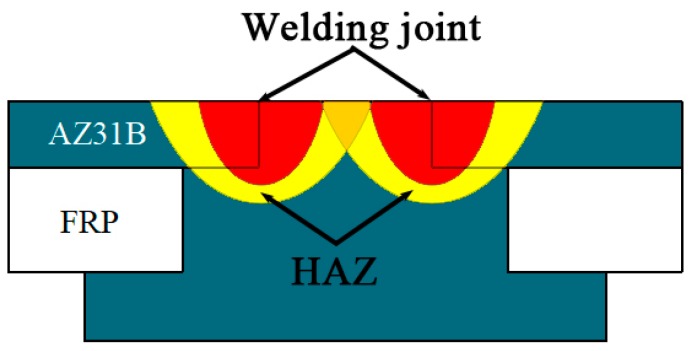
The diagram of the bonding structure of the joint.

**Figure 8 materials-12-02167-f008:**
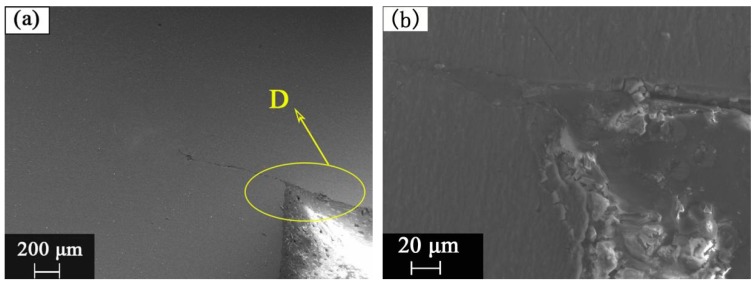
The SEM images of (**a**) A region in Figure 5a and (**b**) D region.

**Figure 9 materials-12-02167-f009:**
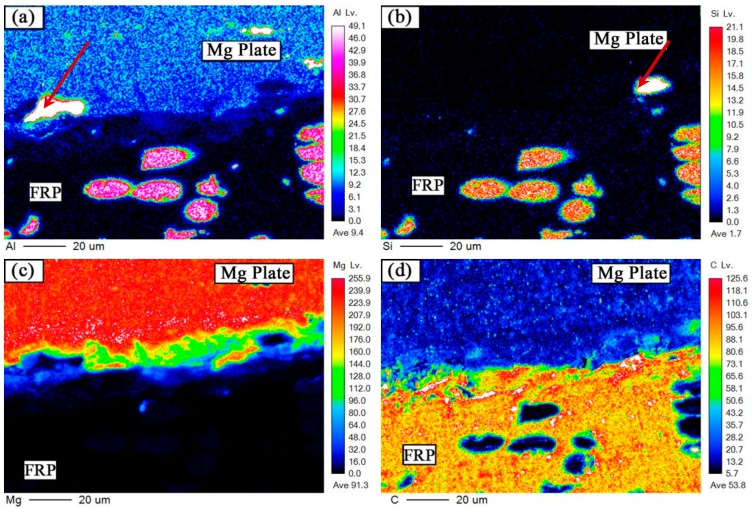
EPMA of the C region in Figure 5d. Elements distribution of (**a**) Al, (**b**) Si, (**c**) Mg, and (**d**) C.

**Figure 10 materials-12-02167-f010:**
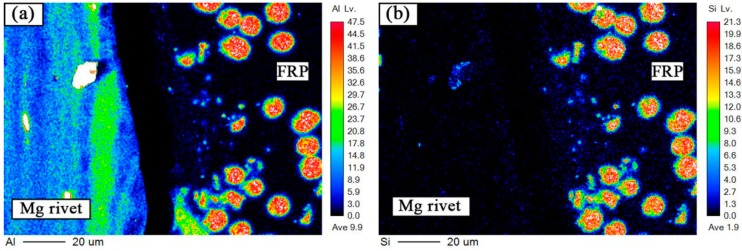
EPMA of the B region in Figure 5d. Elements distribution of (**a**) Al, (**b**) Si, (**c**) Mg, and (**d**) C.

**Figure 11 materials-12-02167-f011:**
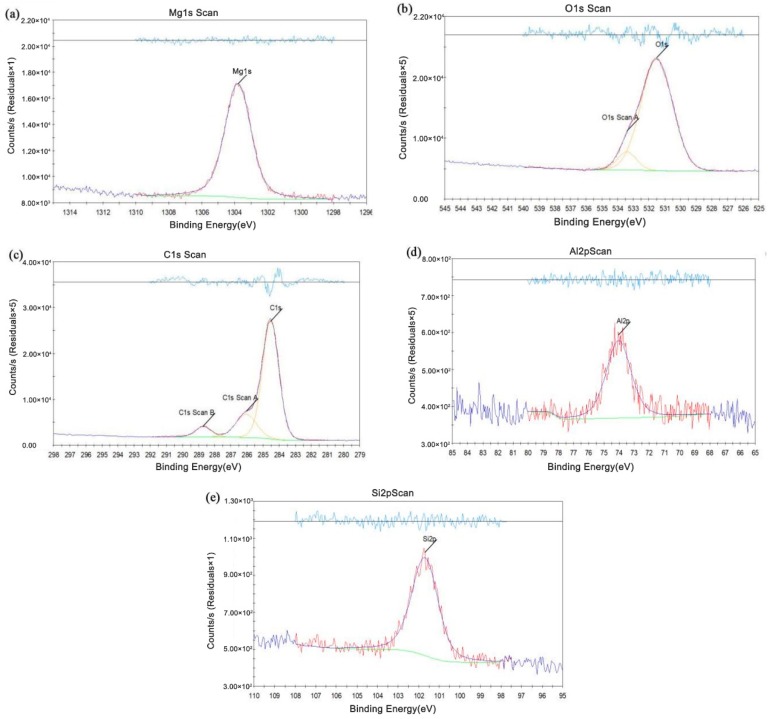
XPS of the interface in Figure 9d: (**a**) Mg, (**b**) O, (**c**) C, (**d**) Al, and (**e**) Si.

**Figure 12 materials-12-02167-f012:**
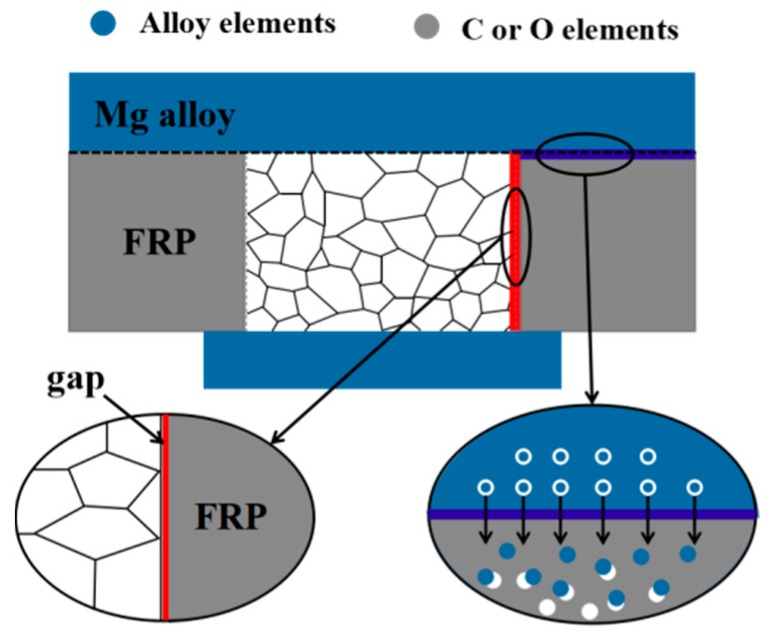
The diagram of the interfaces.

**Table 1 materials-12-02167-t001:** The chemical composition of AZ31B.

Element	Al	Si	Ca	Zn	Mn	Cu	Mg
Quantity	3.2	0.07	0.04	1.2	0.8	0.01	extra

**Table 2 materials-12-02167-t002:** Hybrid welding parameters.

Welding Parameters	Value
Welding speed (rad/min)	36
Arc welding current (A)	70
Laser power (W)	215
Defocus distance (mm)	0

**Table 3 materials-12-02167-t003:** Microstructures of the rivet I joint and the rivet II joint.

Area	The Rivet I Joint	The Rivet II Joint
Fusion zone	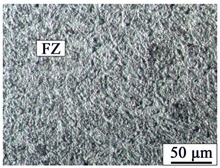	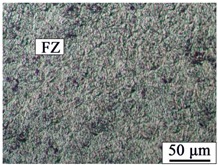
HAZ on Mg plate	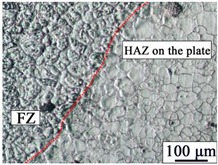	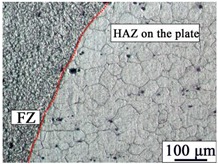
HAZ on rivet	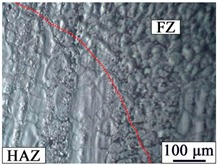	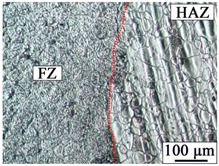

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
