# Peer review of "Investigation of the Joining Technology of FRP/AZ31B Magnesium Alloy by Welding and Riveting Hybrid Bonding Method"

_materials, 2019, doi:10.3390/ma12132167_

Round 1

Reviewer 1 Report

Dear Authors,

I have read Your work with great attention and interests. I’m convinced that the manuscript falls within the scope of the journal and the work is sufficiently original.

The manuscript title is long and twice word: “joining” is included.

In my opinion, in the abstract, keywords and perhaps in the title, there should be information that the research concerns the laser-TIG process.

Introduction is very well written. However, there is no specification of the scope of different hybrid welding processes (you can cite recent (last 2 years) articles from Materials and Metals journals in this topic).

Line 36: “auto native energy”? This is an incomprehensible phrase.

Line 39: from the word "Research ..." there should be a new paragraph.

Line 46: change is to are

Lines 55 and 66: change property to properties.

Line 67: welding is a better word than bonding.

Line 80 and 82: consider changing the word structure to shape.

Fig. 1. photos should be signed as a) and b)

Line 89: the description of the welding procedure should contain all information about the laser-TIG process, eg. the name of the process (it is only in the table), type and flow rate of shielding gas, diameter and type of consumable (is this the 141 or 142 process?), type and diameter of the non-consumable electrode…

Fig. 2. fill the figure caption with a, b, c meanings.

Table 2.: change TIG current to arc welding current, add spaces before units.

Lines 115-117: this information should be provided in earlier.

Line 134: check is the term smoothness used correctly?

Line 217: "Mg native oxide" is very rarely used term.

Line 236: I propose to change well to successfully.

Line 248: There is no comment on the last sentence. Is the formation of oxides a phenomenon that has a positive effect on the properties of joints?

References: format sources acc. to journal guidelines. There is only one article from the Mdpi publishing house.

Editorial remarks:

Line 53: add space before unit. Change Laser to laser.

Line 34: delete space before dot.

Line 43: add space after dot.

Lines 82 and 83: Add spaces before units.

Fig. 3. Add spaces in figure caption.

Author Response

Dear Editor:

Thank you so much for responding to my submission. And I have mend the manuscript according to the reviewer’s suggestions. If you have any problem, you can contract with me by the email.

The answer to the first reviewer

1) The manuscript title is long and twice word: “joining” is included.

Answer: the title is changed as “Investigation of the joining technology of FRP/AZ31B magnesium alloy by welding and riveting hybrid bonding method”

2) In my opinion, in the abstract, keywords and perhaps in the title, there should be information that the research concerns the laser-TIG process.

Answer: it was difficult to add the laser-TIG welding process in the title, and it was add in the abstract and keywords.

3) Introduction is very well written. However, there is no specification of the scope of different hybrid welding processes (you can cite recent (last 2 years) articles from Materials and Metals journals in this topic).

Answer: Some references about the hybrid welding process were added in the manuscript.

4) Line 36: “auto native energy”? This is an incomprehensible phrase.

Answer: it was changed as “new energy vehicles”

5) from the word "Research ..." there should be a new paragraph.

Answer: it was changed as the reviewer’s suggestion.

6) Line 46: change is to are

7) Lines 55 and 66: change property to properties.

8) Line 67: welding is a better word than bonding.

9) Line 80 and 82: consider changing the word structure to shape.

10) Fig. 1. photos should be signed as a) and b)

Answer: They have been mended as the reviewer’s suggestions.

11) Line 89: the description of the welding procedure should contain all information about the laser-TIG process, eg. the name of the process (it is only in the table), type and flow rate of shielding gas, diameter and type of consumable (is this the 141 or 142 process?), type and diameter of the non-consumable electrode…

Answer: The hybrid welding process is added in the manuscript as blow.

The parameters of the pulse laser were as follows: 1.064μm wavelength, 120mm core diameter, 0mm defocusing distance. During the welding experiment, 99.99% pure argon was used as shielding gas from the TIG torch. In the paper a pulsed Nd: YAG laser with a maximum average power of 1000 W and a 70A-ungsten inert gas (TIG) electrical arc were used as hybrid heat sources.

12) Fig. 2. fill the figure caption with a, b, c meanings.

Answer:  It has been added in the manuscript such as: a. prefabricated hole in Mg and FRP, b. laser-TIG hybrid welding process

13) Table 2.: change TIG current to arc welding current, add spaces before units.

14) Lines 115-117: this information should be provided in earlier.

Answer: They were changed as the reviewer’s suggestions.

15) Line 134: check is the term smoothness used correctly?

Answer: The sentence is changed as “The welding appearance of the first joint (rivet I) is wider than the second one (rivet II).”

16) Line 217: "Mg native oxide" is very rarely used term.

Answer: It was changed as Mg oxides.

17) Line 236: I propose to change well to successfully.

Answer: It was changed as the reviewer’s suggestions.

18) Line 248: There is no comment on the last sentence. Is the formation of oxides a phenomenon that has a positive effect on the properties of joints?

Answer: It was changed as “The results of EPMA and XPS showed that there might be Mg oxide, Al oxide and aluminosilicate in the Mg plate/FRP plate interface, which was helpful for the improvement of the dissimilar materials joint.”

Line 53: add space before unit. Change Laser to laser.

Line 34: delete space before dot.

Line 43: add space after dot.

Lines 82 and 83: Add spaces before units.

Fig. 3. Add spaces in figure captio

All above have been mend.

Reviewer 2 Report

Manuscript Ttitle: Investigation of the joining technology of FRP/AZ31B magnesium alloy by welding and riveting hybrid joining method

This manuscript presents an investigation about the strength of a composite riveted to one magnesium alloy and the influence of the welding on the riveted joint.

Various comments will be useful for the authors:

1.      Line 19: "1419N"; Line 64: "200 MPa". Please always use the same rule (space). The best one is the second case. This issue happens several times along the manuscript.

2.      Page 2. Line 76. What FRP was used in the experiments? Characteristics? Fibers type? Strength? Manufacturer? Thermoplastic?

3.      Page 3. Figure 2. Too small letters.

4.      Page 3. Table 2. There is no protective gas used during welding? Which laser was used? CW? Pulsed?

5.      Page 4. Line 121-122. “strain rate of 1.0 mm/min”. The units "mm/min" are for "speed", not for "strain rate".

6.      Page 5. Line 146. “plate with a hole is shown in Fig. 6(a)”. What diameter is?

7.      Page5. Line 150. “It can be inferred that during the stretching process, and the force forms were similar” – Rephrase, please.

8.      Page 5. Lines 146-147. This peak load means that the stress in the plate is 20-30 MPa. This is too low. What composite is this? One CFRP has 1500 MPa strength.

9.      Page 6. Figure 6. Vertical axis: on (a) is only "Tensile load"; in (b-c) is "tensile and shear". Therefore, maybe, the better is to write only "Load (N)".

10.  Page 6. Line 157. The rivet is under a stress of approx. 88/pi()*4=64 MPa in shear and 42 MPa tensile. These stresses are too low for an Mg alloy. That means the weld is not good enough or not well done.

11.  Page 6. Line 159. In English, please.

12.  Page 6. Line 174. "welding beam"?

13.  Page 7. Lines 176-178. The reason for low hardness is not because the welding speed is fast. In fact, for fast speed, there is a high rate decreasing temperature, therefore temper can happen. Did the Author measure the hardness?

14.  Page 8. Figure 8. Looking to Figure 5, this Figure 8 do not appear about the same region.

15.  Page 8. Figure 8. The caption says "Fig. 4". It is not right.

16.  Page 8. Line 201. Instead of "Fig. 4", it is "Fig. 5".

17.  Page 9. Caption of Figure 10. Instead of "Fig. 4", it is "Fig. 5".

18.  In this experiment what is the advantage of Laser-TIG instead oh only Laser?

19.  Why has the Mg alloy so low strength? It was very important to weld Mg to Mg and check the procedure. May be the lack of protection gas is the answer.

Author Response

The answer to the second reviewer

1) Line 19: "1419N"; Line 64: "200 MPa". Please always use the same rule (space). The best one is the second case. This issue happens several times along the manuscript.

Answer: The 200Mpa was a cited from reference, therefore it was hard to know the exactly load of the joint. And only this one was shown in the strength of the joint.

2) Page 2. Line 76. What FRP was used in the experiments? Characteristics? Fibers type? Strength? Manufacturer? Thermoplastic?

Answer: The fiber in the materials is the carbon fiber, and the content of the fiber is only 20%, therefore the property of the FRP is not uniform, which is about 80-120Mpa. The base material of the FRP is the polyether-ether-ketone thermoplastic. And the above information is added in the manuscript.

3) Page 3. Figure 2. Too small letters.

Answer: The figure.2 has been changed.

4) Page 3. Table 2. There is no protective gas used during welding? Which laser was used? CW? Pulsed?

Answer: The parameters of the pulse laser were as follows: 1.064μm wavelength, 120mm core diameter, 0mm defocusing distance. During the welding experiment, 99.99% pure argon was used as shielding gas from the TIG torch. In the paper a pulsed Nd: YAG laser with a maximum average power of 1000 W and a 70A-ungsten inert gas (TIG) electrical arc were used as hybrid heat sources.

5) Page 4. Line 121-122. “strain rate of 1.0 mm/min”. The units "mm/min" are for "speed", not for "strain rate".

Answer: the stain rate is changed as strain speed.

6) Page 5. Line 146. “plate with a hole is shown in Fig. 6(a)”. What diameter is?

Answer: the diameter of the plate is same with hybrid bonding joint, which is 8mm.

7) Page5. Line 150. “It can be inferred that during the stretching process, and the force forms were similar” – Rephrase, please.

Answer: It can be inferred that the force form of rivet I was similar with the rivet II joint during the stretching process

8) Page 5. Lines 146-147. This peak load means that the stress in the plate is 20-30 MPa. This is too low. What composite is this? One CFRP has 1500 MPa strength.

Answer: The fiber in the materials is the carbon fiber, and the content of the fiber is only 20%, therefore the property of the FRP is not uniform, which is about 80-120Mpa. The base material of the FRP is the polyether-ether-ketone thermoplastic. And as there is a hole in the plate, therefore the tensile load of the CFRP is lower.

9) Page 6. Figure 6. Vertical axis: on (a) is only "Tensile load"; in (b-c) is "tensile and shear". Therefore, maybe, the better is to write only "Load (N)".

Answer: The title of the vertical axis in Fig.6 is changed as Load(N).

10)Page 6. Line 157. The rivet is under a stress of approx. 88/pi()*4=64 MPa in shear and 42 MPa tensile. These stresses are too low for an Mg alloy. That means the weld is not good enough or not well done.

Answer: The load of the rivet I joint do not present the property of the Mg welding joint. It shown the load of pulling out the rivet. As the Mg plate is thin, therefore if the welding joint is only the Mg plate and the first step of the Mg rivet, the property of the joint pulling load will be very low. It is mainly because of the hybrid bonding structure.

11.  Page 6. Line 159. In English, please.

Answer: Line 162 It is mainly related to the heat distribution during the welding process. The rivet structure change the heat transfer in the welding process.

12) Page 6. Line 174. "welding beam"?

Answer: The welding beam is changed as welding joint.

13) Page 7. Lines 176-178. The reason for low hardness is not because the welding speed is fast. In fact, for fast speed, there is a high rate decreasing temperature, therefore temper can happen. Did the Author measure the hardness?

Answer: We have measured the hardness of the joint. The lower of the hardness is found on the HAZ but not the fusion area.

14) Page 8. Figure 8. Looking to Figure 5, this Figure 8 do not appear about the same region.

15)  Page 8. Figure 8. The caption says "Fig. 4". It is not right.

16)  Page 8. Line 201. Instead of "Fig. 4", it is "Fig. 5".

17)  Page 9. Caption of Figure 10. Instead of "Fig. 4", it is "Fig. 5".

Answer: They were all mended.

18) In this experiment what is the advantage of Laser-TIG instead oh only Laser?

Answer: Laser welding method is fit for the hybrid bonding joint, and laser-TIG welding joint is even better, which could be used to control the welding penetration and width of the joint. But in this article we do not discussed the influences of hybrid welding parameters, as it is even more complex process.

19) Why has the Mg alloy so low strength? It was very important to weld Mg to Mg and check the procedure. May be the lack of protection gas is the answer.

Answer: We test the only Mg welding joint, which was over 90% percent of the Mg base metal. However, the load of these hybrid joint is not shown the welding strength of the welding joint. It show the pulling load of the rivet. This load is vertical to the Mg plate, and it is not the shear load of the Mg joint. The vertical plasticity of the Mg welding joint is obviously lower than the Mg alloy, therefore the load of the first rivet is low.

Round 2

Reviewer 1 Report

Dear Authors, 

thank you for accepting most of my comments. I have a few remarks again:

add spaces before units! (again)

line 54: change "property of welding joints" to "properties of welded joints"!

line 103: add t in "tungsten".

table 2: add spaces and change "zero" to 0.

Author Response

Thanks very much for the reviewer.

All problems were mended as the reviewer's comments, which was shown in the manuscript.

Reviewer 2 Report

About the answers:

1.      Answer: The 200Mpa was a cited from reference, therefore it was hard to know the exactly load of the joint. And only this one was shown in the strength of the joint.

There was a misunderstood. I wrote about to have always a space between the number and the unit. It is "1419 N", not "1419N"

2.      Answer: the stain rate is changed as strain speed.

There was a misunderstood. The right is "speed", not "strain rate". The units for strain rate are s-1. It is wrong to write "strain speed"

Author Response

Thanks for the reviewer.

All problems were mended as the reviewer’s comments, which were shown in the manuscript.